# Ready-to-Use Therapeutic Foods Fail to Improve Vitamin A and Iron Status Meaningfully during Treatment for Severe Acute Malnutrition in 6–59-Month-old Cambodian Children

**DOI:** 10.3390/nu15040905

**Published:** 2023-02-10

**Authors:** Sanne Sigh, Nanna Roos, Chamnan Chhoun, Arnaud Laillou, Frank T. Wieringa

**Affiliations:** 1Department of Nutrition, Exercise, and Sports, Faculty of Science, University of Copenhagen, Rolighedsvej 26, 1958 Frederiksberg C, Denmark; 2Department of Fisheries Post-Harvest Technologies and Quality Control, Fisheries Administration, 186 Preah Norodom Boulevard, Phnom Penh 12101, Cambodia; 3Nutrition Section, UNICEF West and Central Africa Region, Dakar 29720, Senegal; 4UMR QualiSud, Institut de Recherche Pour le Développement (IRD), 34394 Montpellier, France; 5Qualisud, University of Montpellier, Avignon University, CIRAD, Institut Agro, IRD, Université de la Réunion, 34394 Montpellier, France

**Keywords:** severe acute malnutrition, RUTF, micronutrient, iron, vitamin A, children

## Abstract

Severe acute malnutrition (SAM) remains a global health concern. Studies on the impact of ready-to-use therapeutic foods (RUTFs) on micronutrient status during SAM treatment are almost nonexistent. The objective was to investigate the impact of RUTFs on the iron and vitamin A status of 6–59-month-old children receiving SAM treatment. Biomarkers of vitamin A status (retinol-binding protein, RBP), iron status (ferritin and soluble transferrin receptor, sTfR), and inflammation (C-reactive protein, CRP, and alpha-1 acid glycoprotein, AGP) were collected at admission and discharge (week 8) during an RUTF effectiveness trial. Vitamin A deficiency was defined as RBP <0.70 µmol/L, low body iron as body iron (BI) <0 mg/kg and deficient iron stores as ferritin <12 µg/L. Data were available for 110 and 75 children at admission and discharge, respectively. There was no significant difference in haemoglobin, ferritin, sTfR, BI or RBP concentrations between admission and discharge. At discharge, BI was 0.2 mg/kg lower, and there was a tendency towards a slightly lower RBP concentration, but the prevalence of vitamin A deficiency was low at admission and discharge (6% and 3%, respectively). The small impact of both RUTFs on improving vitamin A and iron status during SAM treatment calls for further research on the bioavailability of micronutrients to enhance the effectiveness of SAM treatment on micronutrient status.

## 1. Introduction

Severe acute malnutrition (SAM) remains a global health concern in many low- and middle-income countries, with almost 20 million children affected annually, the majority living in South and South-East Asia. SAM is defined as severe wasting (weight-for-height <−3 Z-scores) and/or a low mid-upper-arm circumference (MUAC, <11.5 cm) or nutritional oedema regardless of anthropometric indices. SAM contributes significantly to childhood mortality and morbidity [1,2] and is associated with micronutrient deficiencies, anaemia and infections [3,4,5,6,7,8,9]. Micronutrient deficiencies, including iron, zinc and vitamin A, increase mortality risk and contribute to a high prevalence of infection, deprived growth and sub-optimal cognitive development [10]. Moreover, stunting and micronutrient deficiencies have been estimated to cost USD 266 million annually in Cambodia alone [11].

Children with SAM without complications can be treated with high-energy-dense ready-to-use therapeutic foods [12], which supply all the energy and nutrients required to rehabilitate SAM [13,14]. The effectiveness of RUTFs on weight gain and recovery rates has been well-established, although the weight gains reported are often less than the 4 g/kg/day required by the WHO [15,16,17,18,19]. However, studies assessing the effectiveness of RUTFs for SAM treatment in South-East Asia are very limited. 

Anaemia is estimated to affect 53% of children <6 years of age in Cambodia [20]. The most important contributor to anaemia is often assumed to be iron deficiency [21], but in the latest national micronutrient survey in Cambodia, the prevalence of iron deficiency among children <5 years of age was only 9% [20]. However, haemoglobinopathies are another important cause of anaemia, particularly in Asia. The prevalence of haemoglobinopathies is high in Cambodia, affecting half of the population [22,23]. Studies on SAM treatment have previously reported a high prevalence of anaemia at admission, for example, 80% in Burkina Faso [24], up to 95% in India [9,25], and 49% in Malawi among children with SAM without complications [26].

In Cambodia, in the same micronutrient survey, the prevalence of vitamin A deficiency (VAD) among children <5 years of age was 9%, slightly below the threshold for a major public health concern. However, a large proportion of children under 5 years of age (29%) had a marginal vitamin A status (RBP < 1.05 µmol/L) [20]. A VAD prevalence of 41% and 98% was reported among malnourished hospitalised children in Brazil and the Congo, respectively [27]. As RUTFs aim, in addition to contributing to weight gain, to also maintain or improve micronutrient status, RUTFs contain, among others, 10–14 mg of iron and 0.8–1.1 mg of vitamin A per 100 g [28]. However, few reports exist on the impact of RUTFs on micronutrient status during recovery from SAM. As rehabilitation from SAM requires catch-up growth, micronutrient requirements during rehabilitation are higher than those required to maintain optimal growth and development during childhood. One recent study found a need to reconsider the RUTF fortification levels of vitamin A and iron to fully restore the micronutrient status of children treated for SAM [29], and another study recommended increasing vitamin B1 content for therapeutic foods [30].

Hence, there is an important gap in the evidence on the impact of RUTFs on micronutrient status in the treatment of SAM. To address this research gap, we investigated the impact on the iron and vitamin A status of 6–59-month-old children treated for SAM without complications with RUTFs.

## 2. Materials and Methods

### 2.1. Study Design and Setting

This study was a randomised controlled trial, with data on micronutrient status at admission and discharge after 8 weeks. The trial’s main objective was to test the effectiveness of a novel RUTF (NumTrey, a wafer filled with a fish-based paste) against a standard, milk-based RUTF (BP-100^TM^) for treating SAM among 6–59-month-old Cambodian children. The primary outcomes of the SAM project have been reported elsewhere [31,32,33]. Sigh et al. 2018 [31] describe the methodology fully.

The intervention trial was conducted from September 2015 until January 2017 at the National Paediatric Hospital in Phnom Penh, Cambodia. The sample size was calculated based on the superiority of either RUTF for the primary outcome, weight gain (g/kg/day). To detect a 10% difference and a standard deviation of 0.7 g/kg/day, a sample size of 49 children in each group was required to achieve 80% power (α = 0.05, two-sided). To account for attrition, 120 children (60 children/group) were enrolled in the trial [31].

### 2.2. Study Population

As described previously, the eligibility criterion for this study was 6–59-month-old children with non-complicated SAM or borderline SAM, defined as a weight-for-height z-score (WHZ) ≤ −2.8. Children were also enrolled based on MUAC ≤ 115 mm and/or the presence of nutritional oedema [31]. The inclusion criteria deviated from the WHO definition of SAM to ensure that the children treated for SAM as inpatients could continue outpatient treatment through this trial; hence, the cutoff used in the present trial was adjusted.

Children were not eligible if they had any complications affecting food intake or were participating in other clinical trials. Personal and socio-demographic information was obtained from caregivers using questionnaires. The age of the children was obtained from either birth certificates or vaccination and growth-monitoring cards or, if unavailable, by interviewing the caregiver. The children received treatment for SAM for 8 weeks. If the children had not recovered within the trial period, they continued treatment under the national SAM treatment program.

### 2.3. Ready-to-Use Therapeutic Foods and Treatment

A local manufacturer (Vissot/Danish Care Foods) produced the local RUTF (NumTrey) [32], and the standard RUTF (BP-100) was imported by UNICEF (Compact, Oslo, Norway). Table 1 shows the micronutrient compositions of the two RUTFs compared with the UN requirements.

As shown in Table 1, the local RUTF had a slightly lower energy and micronutrient content per 100 g. The wafer wrapping the fish-based paste had a lower nutrient density than the paste. Therefore, during this trial, the children were provided with an RUTF ration, which was calculated based on the nutrition composition of the fish paste without the wafer. This resulted in the distribution of a slightly higher food ration (g/d) to the patients receiving NumTrey. Following the national standards at the time, the food ration distributed to each child was based on the child’s weight, calculated to be between 160 and 180 kcal/kg for both RUTFs [34]. The iron and vitamin A content was calculated based on the daily dose per treatment arm described (Table 2).

### 2.4. Blood Sampling and Laboratory Analyses

At admission (week 0) and discharge (week 8), 4 mL of venous blood was drawn from the cubital vein using a sterile 23G/25G × ¾” scalp vein set (Vihankook) or a sterile 24G IV cannula needle with a catheter and injection valve (HEALFLON^TM^, Harsoria Healthcare Private Limited, Udyog Vihar, India) by trained medical staff. Approximately 2 mL of whole blood was collected in a 6 mL trace-element sodium NH heparin vacutainer (VACUETTE^®^, Greiner Bio-One International GmbH, Frickenhausen, Germany), and the remaining blood was allocated in 2 × 2 mL ethylene diamine tetra acetic vacutainers to analyse haemoglobin concentration (anaemia), haemoglobinopathies and other biomarkers reported elsewhere [33].

At the National Paediatric Hospital, approximately 300 µL of whole blood (EDTA-treated collection tubes) was used to assess haemoglobin concentrations using a haematology analyser (Sysmex Kx-21, Sysmex Corporation, Kobe, Japan, ID. No. NP021 and NP023). The presence of haemoglobinopathies was analysed using electrophoreses (Minicap, SEBIA, Lisses, France) at the National Paediatric Hospital. However, due to a disruption in the capacity to analyse at the hospital, five samples were analysed at the Pasture Institute, Phnom Penh, using the same method and analysis protocol.

The blood samples were stored in a cool box with ice packs while being transported (~30 min) from the hospital to the laboratory at the Department of Fisheries Post-Harvest Technologies and Quality Control for processing. The blood was centrifuged (GlobalRoll^®^; model 80-2B, Zhejiang, China) for 10 min (2700× *g*). The plasma was aliquoted into three tubes, with a 0.2 mL Eppendorf PCR tube used for micronutrient status determination. The plasma was stored at <−20 °C until packed with dry ice and shipped to the VitMin laboratory (Willstaett, Germany) for biochemical analyses of iron status (ferritin (FER) and soluble transferrin receptor (sTfR)), vitamin A status (retinol-binding protein (RBP)) and inflammation (C-reactive protein (CRP) and α-1-acid- glycoprotein (AGP)). A combined sandwich ELISA method in duplicate was used to analyse FER, sTfR, RBP, CRP and AGP [35].

### 2.5. Definition of Micronutrient and Health Status

Haemoglobin concentrations were used to diagnose anaemia status using cutoff values recommended by the WHO [36]. A normal haemoglobin genetic pattern was classified as HbA > 70%, HbF ≤ 30% and HbA2 < 2%, and HbA > 95.5%, HbA2 2.0–3.5% and HbF < 5% for 0–23-month-old children and 24–59-month-old children, respectively. All remaining haemoglobin variations were classified as haemoglobinopathies, except for HbE-Heterozygote (HbE between 20 and 30%), which was classified as a separate entity, given the high prevalence [23].

Inflammation was categorised into four groups, that is, healthy and three stages of inflammation, according to Thurnham et al. [37,38]. The inflammation stages were incubation (elevated CRP > 5 mg/L and normal AGP), early convalescence (elevated CRP > 5 mg/L and AGP > 1 g/L) and late convalescence (normal CRP and elevated AGP > 1 g/L). Ferritin and RBP concentrations were corrected for inflammation using the correction factors derived by Thurnham et al. The ferritin and RBP concentrations of the children without inflammation were not adjusted. Based on a meta-analysis by Turnham et al. 2010 [37], the FER levels were multiplied by 0.77, 0.53 and 0.75 according to the inflammation group, incubation, early convalescence, and late convalescence, respectively. RBP was corrected by factors of 1.15, 1.1.32 and 1.12 for the incubation, early convalescent and late convalescent groups, respectively [38].

No iron stores were defined as FER < 12 μL for children <60 months of age, hence indicating iron deficiency [23]. The cutoff for vitamin A deficiency was defined as an RBP < 0.70 μmol/L [38]. Marginal vitamin A status was defined as RBP 0.70–1.05 μmol/L [39], which is linked with liver stores of retinol [40].

Body iron (BI) was calculated by using the methods devised by Cook et al. [41], using adjusted ferritin concentrations, as follows: Body iron mg/kg = −[log10((sTfR) × 1000/adjusted ferritin) −2.8229]/0.1207. A low BI was defined as BIS < 0 mg/kg [22].

### 2.6. Ethical Considerations

The study protocol was approved by the National Ethical Committee for Health Research of the Ministry of Health, Kingdom of Cambodia (April 2015 Version No. 2). The study was conducted according to the guidelines of the Declaration of Helsinki. Both written and verbal informed consent were obtained from all caregivers. The trial was registered at ClinicalTrials.gov (name: “Comparison of a Locally Produced RUTF With a Commercial RUTF in the Treatment of SAM (FLNS_SAM)”; Trial registration: NCT02907424; URL: https://clinicaltrials.gov/ct2/show/NCT02907424 (accessed on 15 December 2022)).

### 2.7. Statistical Analyses

All data were double-entered into EpiData version 3.1 (The EpiData Association, Odense, Denmark). Statistical analyses were conducted using R studio software (Version 3.4.4, Windows) and SPSS (Version24, IBM). A visual assessment of residuals and a normal probability plot inspection were conducted for all models. Non-normal distributed models were log^10^-transformed.

The descriptive statistics are presented as percentages (*n*) or means (standard deviation, SD). The difference between admission and discharge was analysed using a paired *t*-test (numeric variables) or Fisher’s exact test (categorical variables). The comparison between the two RUTFs from admission to discharge was analysed using general linear modelling (GLM), adjusted for admission values, age and gender with the Emmeans R package, which considers multiple comparisons. The McNemar test was used to compare the statistical differences in the prevalence of deficiency from admission to discharge. A post hoc analysis was used for the children with a high compliance ≥50%. The analyses are reported as means (SD) and *p*-values. Statistical significance was defined as *p* < 0.05.

## 3. Results

### 3.1. Participants’ Characteristics

A total of 121 children were enrolled in the trial. Data were available for 114 and 75 children at admission and discharge, respectively (Figure 1).

The mean age of the children was 21.5 months, and approximately 60% were males (Table 3). Approximately 57% of the children were anaemic, and 37% had a haemoglobin disorder. Of all children, 12.3% and 3.8% had no iron stores and negative body iron, respectively. In comparison, 31.8% had a marginal vitamin A status, and 5.5% were deficient in vitamin A. Inflammation was present among 16% and 39% based on CRP and AGP, respectively, with an overall 40% having inflammation at baseline. The prevalence of anaemia, haemoglobinopathies, iron deficiency, low body iron and vitamin A deficiency was higher among the children randomised to the standard RUTF than among those randomised to the local RUTF. Contrarily, the proportion of inflammation based on CRP was higher among the children randomised to the local RUTF (19%) than among those randomised to the standard RUTF (12%). Two children allocated to the standard RUTF unfortunately passed away during the study because of progressed HIV infection (one child) and pulmonary tuberculosis (second child) (Figure 1).

### 3.2. Effectiveness of SAM Treatment on Micronutrient Status

There was no significant change in haemoglobin, ferritin or sTfR concentrations between admission and discharge, with the concentrations of the biomarkers for micronutrient status similar to or even lower at discharge than at admission. In a sub-analysis of the children with a high compliance with the RUTF treatment (>50%, *n* = 31), no significant differences in biomarker concentrations between admission and discharge were observed, but the overall trend was towards a better micronutrient status.

There were no significant differences in the prevalence of iron or vitamin A deficiency or anaemia between admission and discharge after 8 weeks of treatment. The prevalence of children with a marginal vitamin A deficiency slightly increased from 31.8% to 32.9% from admission to discharge (*p* = 0.065) (Table 4). The prevalence of iron deficiency was reduced by 6 percent in the small group of children with a high compliance with the treatment, but no significant difference was seen (*p* = 0.63, Table 5).

### 3.3. Comparison between RUTFs

As there were minor differences in the composition, including in the iron and vitamin A content, of the two RUTFs, vitamin A and iron status changes over the 8 weeks of treatment were compared between the two RUTFs (Table 6). No statistical difference in the changes in micronutrient status was found between the two products; therefore, the results for the two RUTFs were combined.

## 4. Discussion

In this study, we aimed to assess the impact of RUTFs on the micronutrient status of children with severe acute malnutrition. The prevalence of anaemia and the deficiencies of iron and vitamin A were similar to those reported in the national micronutrient survey but lower than those expected in children with severe acute malnutrition. Indeed, the same pattern of a relatively low prevalence of iron and vitamin A deficiency combined with a high prevalence of anaemia, as found in the national micronutrient survey [20], was present in our study population. Surprisingly, neither iron nor vitamin A status improved significantly over the 8-week treatment. To ascertain that low compliance with the treatment did not underlie this lack of impact, we conducted a sub-group analysis in children consuming >50% of the provided RUTF. However, this sub-group analysis yielded similar results, with only minor shifts in micronutrient status between admission and discharge from treatment. The prevalence of anaemia remained high, above 50%, and it hardly changed over the 2-month treatment (from 57% to 52%, *p* = 0.55). Iron deficiency could only partly explain the high anaemia prevalence, as only 10% of the children had IDA at admission. A study conducted among children receiving treatment for SAM in Burkina Faso observed that anaemia decreased significantly from 78% to 56% from admission to discharge [29]. Similarly, a study with a 12-week treatment of moderate acute malnutrition found a reduction in anaemia from 70% at admission to 53% at discharge [42]. Anaemia has a multi-factorial causality; besides iron, many other factors are involved, both nutritional and non-nutritional. RUTFs supply an extensive range of vitamins and minerals, including those implicated in the aetiology of anaemia (e.g., vitamins A, B2, B6, B9, B12, and copper and zinc) [43,44,45,46,47]. Hence, it could be assumed that all forms of nutritional anaemia would be addressed during treatment with RUTF. The lack of impact on the haemoglobin concentrations in the Cambodian children could thus be due to the fact that non-nutritional causes, such as haemoglobinopathies, were the main cause of anaemia in the present study [22].

During the present study, the compliance with RUTF intake was lower than expected and desired [31], which influenced the intake of micronutrients and, hence, the possible increase in micronutrient status between admission and discharge. In an ad hoc analysis of the data obtained from the present study (data not presented), even in the small sample of children with a high compliance, no indication or significant impact on micronutrient status was found. Hence, it appears that the RUTFs provided insufficient amounts of micronutrients to improve micronutrient status, confirming the similar observations made by Kangas et al. [29]. One possible explanation could be that the bioavailability of iron and vitamin A is low due to inflammation in the gastrointestinal tract, which markers of systemic inflammation, such as CRP and AGP, might not reflect. Small intestinal inflammation (environmental enteric dysfunction and environmental enteropathy) is widespread in low- and middle-income countries and strongly associated with malnutrition, for example, stunting and wasting [48,49,50,51,52], and, hence, environmental enteric dysfunction and enteropathy are likely to influence and limit the effectiveness of fortified foods, such as RUTFs, in improving micronutrient status in malnourished children.

Both the local and standard RUTFs contain ~2 mg more zinc than iron, so it could be speculated that the low increase in iron in this trial was caused by micronutrient interactions between the different nutrients resulting in an inhibited uptake of certain nutrients; for example, zinc and iron compete during intestinal absorption when they are ingested simultaneously, which is the case when children are treated with RUTFs [53]. An excess zinc intake has been found to inhibit zinc absorption and vice versa [54,55]. However, the evidence is conflicting, as well as the discussions on the definitions of interactions between micronutrients [56].

The WHO states that 6–59-month-old children with SAM do not require a high dose of vitamin A supplements, as they receive RUTFs that already contain sufficient vitamin A [12]. This study indicates that vitamin A intake during the RUTF rehabilitation of SAM may be insufficient with the current levels of fortification of the RUTF products.

In particular, iron has been tested in alternative RUTF formulations containing more iron than standard RUTFs. Administering alternative RUTFs to children containing higher amounts of iron (31–35 mg/100 g) compared with standard RUTFs (11 g/100 g) resulted in significantly lower rates of anaemia among children treated with the alternative RUTFs (12–18%) compared to standard RUTF (25%) at discharge [26]. This may indicate that children in countries with high anaemia and iron deficiency rates might benefit from RUTF formulations with a higher iron content. However, in the context of the present study in Cambodia, the higher iron content of the RUTFs would probably have made little difference, given our subjects’ low prevalence of iron deficiency. Moreover, increasing iron intake in SAM requires careful consideration, as several issues may arise, including the negative effects of high iron intake on the human microbiome. Hence, increasing iron content in RUTFs might not apply to countries such as Cambodia, where genetically inherited haemoglobinopathies are common among its population [20]. Another concern with iron is that iron is a pro-oxidative element, which can have negative effects on biological systems at even moderate dosages; for example, beyond altering the gut microbiota, there are some reports on decreased growth (linear and weight); increased illness; and, as mentioned earlier, interactions with other trace elements, such as zinc and copper, and impaired cognitive and motor development [57].

Studies reporting changes in vitamin A status during the RUTF treatment of children with SAM without complications are scarce. There was no impact on circulating RBP concentrations in the present study, although the prevalence of vitamin A deficiency decreased from 6% at admission to 3% at discharge. The prevalence of vitamin A deficiency was considerably higher in a study from Burkina Faso, where a decrease in VAD from 25% at admission to 9% at discharge was found [29]. A National survey in Cambodia conducted a year previous to the study showed that 9% of <5-year-old children suffered from VAD [20]. However, the study did not appear to adjust serum retinol for inflammation. It may have overestimated VAD prevalence [58]. RBP is depressed during infection and inflammation, as it is a negative acute-phase reactant [58].

Unfortunately, although RBP or circulating retinol concentrations do indicate deficiency (<0.70 μmol/L) and marginal vitamin A status (<1.05 μmol/L), they do not reflect the true vitamin A status at higher concentrations. Liver vitamin A concentration is considered the golden standard for vitamin A status because retinol concentration is under homeostatic control to protect humans against toxic concentrations and is used for growth and cellular differentiation in humans.

### Strengths and Limitations

Compliance with RUTF intake was generally low during the effectiveness study. As the study ended after 8 weeks of treatment, not all children might have fully recovered at the time of discharge. The children continued the treatment following the national protocol, but data and biomarkers were not collected after week 8. At the endline, data from 35 children were not collected for various reasons but mainly due to non-compliance with the treatment. Two children passed away due to pre-existing illnesses. Others dropped out or provided insufficient blood collection. As severely ill children are likely to have a worse micronutrient status, discharge data might underestimate the deficiencies at discharge, representing a “best-case” scenario. Moreover, we did not collect data on the provision of high-dose vitamin A capsules (HDVACs) among the patients beyond the treatment received at the hospital. Although the use of HDVACs is limited in Cambodia, we cannot exclude the possibility that some children might have received an HDVAC shortly before admission into the study. Finally, the content of vitamin A within the RUTFs could have decreased during storage, as vitamin A can be unstable, thereby reducing the impact on vitamin A status. However, vitamin A appeared to be relatively stable in this study [59]. It is important to note that the study presents secondary outcomes, so the sample size calculation was not calculated to assess differences in micronutrient status.

Despite these limitations, the study successfully obtained data for biomarkers of a large majority of the children, which allows for a confident estimation of iron status and vitamin A at admission, discharge and change between the two assessment points.

## 5. Conclusions

In conclusion, this study found that both standard and locally produced RUTFs failed to have a meaningful impact on anaemia prevalence and iron and vitamin A status, despite providing a fully fortified food product, and micronutrient status remained sub-optimal after 8 weeks of SAM treatment. The RUTFs may prevent children with SAM from becoming vitamin A deficient to some extent. The little impact of both RUTFs in improving iron and vitamin A during the 8 weeks of SAM treatment calls for further research on how to improve the effectiveness of SAM treatment on micronutrient status to ensure that vulnerable children receive all essential nutrients during treatment.

## Figures and Tables

**Figure 1 nutrients-15-00905-f001:**
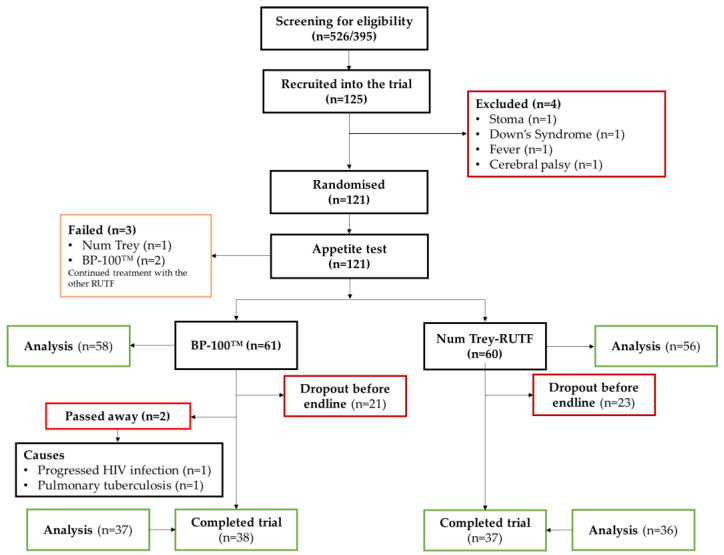
Flowchart of children enrolment into the trial.

**Table 1 nutrients-15-00905-t001:** Micronutrient composition of RUTF per 100 g.

Product Type	Standard RUTF	Local RUTF	UN Requirements
Energy (Kcal)	526	506	520–550
Vitamin A (mg)	0.9	0.8	0.8–1.1
Vitamin D (µg)	18	11.7	15–20
Vitamin E (mg)	27	13.3	≥20
Vitamin K (µg)	21	17.4	15–30
Thiamine (vitamin B1) (mg)	0.5	0.4	≥0.5
Riboflavin (vitamin B2) (mg)	1.8	1.1	≥1.6
Vitamin C (mg)	54	39.3	≥50
Vitamin B6 (mg)	0.7	0.5	≥0.6
Cobalamin (vitamin B12) (µg)	1.6	1.1	≥1.6
Folic acid (µg)	225	249	≥200
Niacin (mg)	5.8	4.1	≥5
Pantothenic acid (mg)	3	2.7	≥3
Biotin (vitamin B7) (µg)	70	94.5	≥60
Calcium (mg)	470	219	300–600
Sodium (mg)	<290	8.26	≤290
Potassium (mg)	1100	773	1100–1400
Phosphorus (mg)	470	297	300–600
Magnesium (mg)	110	88	80–140
Iron (mg)	10	5.8	10–14
Zinc (mg)	12	7.6	11–14
Copper (mg)	1.5	1.1	1.4–1.8
Selenium (µg)	25	20.5	20–40
Iodine (µg)	110	79.0	70–140

Abbreviations: RUTF, ready-to-use therapeutic food; kcal, calories; g, gram; mg, milligrams; µg, microgram.

**Table 2 nutrients-15-00905-t002:** Daily food ration and iron and vitamin A content per RUTF by child’s weight.

Weight (kg)		Standard RUTF				Local RUTF		
	RUTF Quantity/Day (bar)	RUTF Quantity/Day (gram)	Iron Daily RUTF Dose (mg)	Vitamin A Daily RUTF Dose (mg)	RUTF Quantity/Day (Wafer)	RUTF Quantity/Day (gram)	Iron Daily RUTF Dose (mg)	Vitamin A Daily RUTF Dose (mg)
3.0–3.4	2	~114	11.4	1.03	17	~170	9.8	1.3
3.5–4.9	2.5	~142.5	14.3	1.3	20	~200	11.5	1.5
5.0–6.9	4	~228	22.8	2.1	27	~270	15.5	2.1
7.0–9.9	5	~285	28.5	2.6	40	~400	23.0	3.1
10.0–14.0	6	~342	34.2	3.1	53	~530	30.4	4.1

Abbreviations: RUTF, ready-to-use therapeutic food; kg, kilogram.

**Table 3 nutrients-15-00905-t003:** Admission characteristics of children in each group.

	Standard RUTF (*n* = 58)	Local RUTF (*n* = 56)	All Children (*n* = 114)
Socio-demographic parameters			
Age, months	20.5 (12.6)	22.6 (14.7)	21.5 (13.6)
Gender			
Female, % (*n*)	36.2 (21)	44.6 (25)	40.4 (46)
Male, % (*n*)	63.9 (37)	55.4 (31)	59.6 (68)
Anthropometrics			
Weight, kg	7.3 9 (1.6)	7.7 (1.9)	7.51 (1.8)
Height, cm	75.1 (9.4)	77.1 (10.2)	76.1 (9.8)
MUAC, mm	11.8 (0.9)	11.9 (0.7)	11.9 (0.8)
Weight-for-height, z-score	−3.1 (0.7)	−3.0 9 (0.5)	−3.0 (0.6)
Weight-for-age, z-score	−3.4 (0.9)	−3.3 (0.8)	−3.3 (0.8)
Height-for-age, z-score	−2.4 (1;3)	−2.2 (1.5)	−2.3 (1.4)
Anaemia			
Haemoglobin, g/dl	10.7 (1.7)	10.8 (1.3)	10.8 (1.5)
Anaemia, % (*n*) ^1,^*	62.3 (33)	51.0 (25)	56.9 (58)
Haemoglobinopathy **			
Hb normal, % (*n*)	58.2 (32)	71.4 (35)	64.4 (67)
Hb disorders, % (*n*)	42.8 (23)	28.6 (14)	35.6 (37)
HbE-Heterozygote, % (*n*)	27.3 (15)	14.3 (7)	21.2 (22)
HbE-Heterozygote/β-thalassemia, % (*n*)	3.6 (2)	4.1 (2)	3.9 (4)
HbE-Homozygote, % (*n*)	5.5 (3)	4.1 (1)	4.8 (5)
β-thalassemia major, % (*n*)	1.8 (1)	0.0 (0)	1.0 (1)
α-thalassemia, % (*n*)	0.0 (0)	2.0 (1)	1.0 (1)
α-thalassemia/Harbour spring, % (*n*)	0.0 (0)	2.0 (1)	1.0 (1)
Hb-Other, % (*n*)	3.6 (2)	2.0 (1)	3.0 (3)
Iron status *			
Ferritin, µg/L	40.3 (30.0)	41.4 (28.1)	40.8 (28.9)
sTfR, mg/L	10.2 (7.4)	10.1 (9.0)	10.1 (8.2)
BI, mg/kg	3.5 (4.6)	4.1. (3.8)	3.8 (4.2)
Iron deficiency, % (*n*)	19.3 (11)	5.7 (3)	12.7 (14)
Low BI, % (*n*)	17.5 (10)	11.3 (6)	14.5 (16)
Vitamin A status *			
RBP, µmol/L	1.2 (0.5)	1.4 (0.6)	1.3 (0.6)
Marginal vitamin A status, % (*n*)	38.6 (22)	24.5 (13)	31.8 (35)
Vitamin A deficiency, % (*n*)	7 (4)	3.8 (2)	5.5 (6)
Inflammation status *			
CRP, mg/L	4.0 (10.8)	3.1 (5.4)	3.6 (8.1)
CRP, >5 mg/L, % (*n*)	12.3 (7)	18.9 (10)	15.5 (17)
AGP, g/L	1.1 (0.9)	1.0 (0.6)	1.1 (0.8)
AGP, >1 g/L, % (*n*)	38.6 (22)	39.6 (21)	39.1 (43)

Data are reported as percentages (n) or means (SD). *^1^* Anaemia cutoff is age- and sex-dependent. Abbreviations: MUAC, mid-upper-arm circumference; sTfR, soluble transferrin receptor; BI, body store; CRP, C-reactive protein; AGP, α_1_-acid glycoprotein. * No micronutrient data for one child (standard RUTF) and three children (local RUTF), hence a total of 4 children. ** Missing data from 10 children (*n* = 3 standard RUTF, and *n* = 7 local RUTF).

**Table 4 nutrients-15-00905-t004:** Vitamin, mineral and inflammation status during the SAM effectiveness trial.

	6–59-Month-Old Children Admission Mean Age of 21.5 Months				
Biomarker	Admission (*n* = 110)		Discharge (*n* = 75)		*p*-Value ^1^
	Mean	SD	Mean	SD	
Haemoglobin, g/dl	10.8	1.5	11.0	1.6	0.27
Ferritin, µg/L	40.8	28.9	37.0	23.6	0.74
sTfR, mg/L	10.1	8.2	9.8	8.6	0.46
BI, mg/kg	3.8	4.2	3.6	3.8	0.90
RBP, µmol/L	1.3	0.6	1.2	0.6	0.36
CRP, mg/L	3.6	8.6	2.6	5.7	0.39
AGP, g/L	0.96	0.69	0.79	0.57	0.08
Iron deficiency, %	12.5		9.6		1.00
Low BI, %	14.5		16.4		0.73
Marginal vitamin A, %	31.8		32.9		0.65
Vitamin A deficiency, %	5.5		2.7		1.00
Anaemia, %	56.9		52.0		0.55

Abbreviations: sTfR, soluble transferrin receptor; BI, body iron; RBP, retinol-binding protein; CRP, C-reactive protein; AGP, α_1_-glycoprotein; SE, standard error. *p* < 0.05 is considered a statistically significant difference. ^1^
*p*-values are for comparing admission to discharge with GLM repeated measures variables and McNemar for deficiency.

**Table 5 nutrients-15-00905-t005:** Vitamin, mineral and inflammation status during the SAM effectiveness trial among children with (>50%) high compliance.

6–59-Month-Old Children Admission Mean Age of 19.7 Months					
Biomarker	Admission (*n* = 31)		Discharge (*n* = 31)		*p*-Value ^1^
	Mean	SD	Mean	SD	
Haemoglobin, g/dl	10.7	1.3	10.9	1.2	0.25
Ferritin, µg/L	32.0	27.4	33.5	19.9	0.75
sTfR, mg/L	12.2	10.7	10.1	8.8	0.16
BI, mg/kg	2.3	5.0	3.3	4.1	0.19
RBP, µmol/L	1.24	0.44	1.33	0.77	0.55
CRP, mg/L	4.3	10.7	3.5	7.9	0.74
AGP, g/L	1.0	0.74	0.9	0.69	0.69
Iron deficiency, %	19.4		12.9		0.63
Low BI, %	22.6		19.4		1.00
Marginal vitamin A, %	32.3		22.5		0.51
Vitamin A deficiency, %	6.4		3.2		1.00
Anaemia, %	56.7		54.8		1.00

Abbreviations: sTfR, soluble transferrin receptor; BI, body store; RBP, retinol-binding protein; CRP, C-reactive protein; AGP, α_1_-glycoprotein; SE, standard error. *p* < 0.05 is considered a statistically significant difference. ^1^
*p*-values are for comparing admission to discharge with GLM repeated measures variables and McNemar for deficiency.

**Table 6 nutrients-15-00905-t006:** Estimated micronutrient and inflammation status at discharge from SAM treatment per RUTF.

Biomarker	Standard RUTF (*n* = 36)		Local RUTF (*n* = 33)		Difference between Treatments (Compared to Control)		
	Mean	SD	Mean	SD	Estimated Difference	SE	*p*-Value
Haemoglobin, g/dl	11.4	1.8	10.5	1.1	−0.58	0.35	0.10
Ferritin, µg/L	36.5	24.1	37.2	24.7	−0.20	5.4	0.98
sTfR, mg/L	9.7	7.2	10.3	10.4	−0.04	2.1	0.99
BI, mg/kg	3.4	4.0	3.6	3.9	+0.49	0.93	0.60
RBP, µmol/L	1.23	0.75	1.28	0.37	+0.06	0.11	0.58

Abbreviations: sTfR, soluble transferrin receptor; BI, body iron; RBP, retinol-binding protein; CRP, C-reactive protein; AGP, α_1_-glycoprotein; SE, standard error. *p* < 0.05 is considered a statistically significant difference.

## Data Availability

The data presented in this study are available on request from the corresponding author.

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
