# Peer review of "Ready-to-Use Therapeutic Foods Fail to Improve Vitamin A and Iron Status Meaningfully during Treatment for Severe Acute Malnutrition in 6–59-Month-old Cambodian Children"

_nutrients, 2023, doi:10.3390/nu15040905_

Round 1

Reviewer 1 Report

Please provided the reason for deviation from the WHO definition of children with SAM (<-3 SD).

What is the source of the calorie recommendation (160-180 kcal/kg) used for food ration distribution?

What percentage of the recommended requirements did the nutrients (iron and Vit A) meet for children with SAM?

What was the exact quantity (in grams) of RUTF (per arm) distributed?

Provide anthropometry details in Table 3.

How was iron deficiency defined?

Most of the variables (iron indices) seem not normally distributed, consider providing median and quartiles.

Is it possible that nutrient interactions (for example, between iron and zinc) or malnutrition enteropathy reduced the effect of the RUTF? Add in discussion.

Iron intake has shown to be detrimental for growth (Bo Lönnerdal, Excess iron intake as a factor in growth, infections, and development of infants and young children, The American Journal of Clinical Nutrition, Volume 106, Issue suppl_6, December 2017, Pages 1681S–1687S), should we be cautious in administering iron in iron replete children. Discuss.

Add sample size as a limitation as the original sample size calculation was not targeted to the present study outcome. 

Minor

1. Table 2: Food ‘ratio’ to ‘ration’

2. Line 193 close brackets for SD.

3. Add reference for lines 317 – 321.

Author Response

Reviewers' report, revisions and response to reviewers' reports                                                         

Dear Reviewers,

A sincere thanks for spending your valuable time reviewing our manuscript. The concerns have been addressed in reply to your comments below. The comments and suggestions have been ordered as R1 (reviewer 1) and C# (comment no) (example R1.C1). The revisions in the text are marked using the" Track Changes" function in MS Word.

Reviewer 1's report and authors' comments/revisions

Open Review

English language and style

( ) English very difficult to understand/incomprehensible
( ) Extensive editing of English language and style required
( ) Moderate English changes required
(x) English language and style are fine/minor spell check required
( ) I don't feel qualified to judge about the English language and style

Yes

Can be improved

Must be improved

Not applicable

Does the introduction provide sufficient background and include all relevant references?

(x)

( )

( )

( )

Are all the cited references relevant to the research?

(x)

( )

( )

( )

Is the research design appropriate?

( )

( )

( )

(x)

Are the methods adequately described?

( )

( )

(x)

( )

Are the results clearly presented?

( )

( )

(x)

( )

Are the conclusions supported by the results?

( )

( )

(x)

( )

Reviewer comments to Authors:

R1.C1: Please provided the reason for deviation from the WHO definition of children with SAM (<-3 SD).

R1.C1 Authors reply: The children recruited into our study were all diagnosed with severe acute malnutrition (SAM), and had either started their SAM treatment as an inpatient, where complications were treated before discharge, and continuation of SAM treatment as an outpatient, or were directly referred to outpatient treatment of SAM. In general, trials on community-based management of acute malnutrition (CMAM) could either receive patients directly referred to CMAM, or receive patients treated as an inpatient prior to inclusion to home-based treatment, the latter resulting in some patients bordering between SAM and moderate acute malnutrition (MAM) at baseline of CMAM trials, even though they were SAM at the start of the inpatient treatment. The cutoff used as inclusion criteria was adjusted to ensure these patients were treated inpatient before trial enrolment. A detailed explanation was included in the article reporting on the primary results of the trial: https://www.mdpi.com/2072-6643/10/7/909. .

For clarification, the following was added to section 2.2. Study population line 100: Children were also enrolled based on MUAC ≤115 mm and/or the presence of nutritional oedema [31]. The inclusion criteria deviated from the WHO definition of SAM to ensure that children treated for SAM as inpatient could continue outpatient treatment through this trial; hence, the cut-off used in the present trial was adjusted."

R1.C2: What is the source of the calorie recommendation (160-180 kcal/kg) used for food ration distribution?

R1.C2 Authors reply: WHO recommends between 150 and 220 kcal/kg/day (https://apps.who.int/iris/bitstream/handle/10665/352673/9789240029941-eng.pdf?sequence=1&isAllowed=y). At the time of the study the food ration followed national (Cambodian) food rations standards with was between 160-180 kcal/kg/day.

For clarity, the following, including a reference, was added to lines 121-123: "receiving NumTrey. Following national standards at the time, the food ration distributed to each child was based on the child's weight, calculated to be between 160-180 kcal/kg for both RUTF [34]."

R1.C3: What percentage of the recommended requirements did the nutrients (iron and Vit A) meet for children with SAM?

R1.C3 Authors reply:

Recommended daily requirements (e.g. RDA) were developed for healthy people, and reflect long-term intakes. Children with SAM are required to have higher intakes, due to the need to lay down tissue rapidly.

For children with SAM, the RUTF is intended to provide all the daily recommended requirements, and therefore international requirements for RUTF have a higher vitamin A and iron density. For iron and vitamin A, the recommended quantities per 100 g of RUTF are 10-14 mg iron/100 g RUTF and 0.8-1.1 mg vitamin A/100 g RUTF respectively. Both RUTFs used in our study fulfilled these criteria for vitamin A, but the local RUTF was low on iron (5.8 mg instead of 10 mg).

For vitamin A, the RDI for children 1 – 8 years of age is in the order of 0.4 – 0.5 mg retinol equivalent/day, while the upper limit is in the range of 0.6 – 0.9 mg/day. These recommendations are for long-term intake, and for healthy children. In our study, 100 g of RUTF provided around 2x the RDI for vitamin A for children 1 – 4 years of age.

For iron, 100 g of RUTF provided 10 or 5.8 mg of iron. The RDA for healthy children 1 – 4 years of age is in the order of 7 to 10 mg of iron/day. Hence 100 g of RUTF provided 1 RDA or 0.6 RDA of iron respectively. The upper limit for iron is 40 mg/d for children 1 – 13 years of age.

Table 1 presents the required nutrient compositions of the RUTF per 100g, including the two RUTFs used in the present trial. RUTFs should be able to cover the nutrient needs with daily rations given 150-220/kg/day (as defined by WHO). Despite the local RUTF having lower (almost 50%) iron content, no statistically significant changes were found on the marker of iron at discharge.

R1.C4: What was the exact quantity (in grams) of RUTF (per arm) distributed?

R1.C4 Authors reply: Table 2 has been adjusted the reflect the grams of RUTF distributed based on the children's weight of children 6-59-months old.

Revised table:

Table 2. Daily food ration and iron and vitamin A content per RUTFs by child's weight.

Weight (kg)

Standard RUTF

Locally RUTF

RUTF quantity/day (bar)

RUTF quantity/day (gram)

Iron daily RUTF dose (mg)

Vitamin A daily RUTF dose (mg)

RUTF quantity/day (wafer)

RUTF quantity/day (gram)

Iron daily RUTF dose (mg)

Vitamin A daily RUTF dose (mg)

3.0-3.4

2

~114

11.4

1.03

17

~170

9.8

1.3

3.5-4.9

2.5

~142.5

14.3

1.3

20

~200

11.5

1.5

5.0-6.9

4

~228

22.8

2.1

27

~270

15.5

2.1

7.0-9.9

5

~285

28.5

2.6

40

~400

23.0

3.1

10.0-14.0

6

~342

34.2

3.1

53

~530

30.4

4.1

Abbreviations; RUTF, ready-to-use therapeutic foods; kcal, calories; kg, kilogram.

R1.C5: Provide anthropometry details in Table 3.

R1.C5 Authors reply: weight (kg), height (cm), MUAC (mm), weight-for-height z-score, weight-for-age z-score, and height-for-age z-score have been added to Table 3.

Table 3: Admission characteristics of children in each group.

Standard RUTF

(n = 58)

Local RUTF

(n = 56)

All children

(n = 114)

Socio-demographic parameters

Age, months

20.5 (12.6)

22.6 (14.7)

21.5 (13.6)

Gender

Female, % (n)

36.2 (21)

44.6 (25)

40.4 (46)

Male, % (n)

63.9 (37)

55.4 (31)

59.6 (68)

Anthropometrics

    Weight, kg

7.3 9 (1.6)

7.7 (1.9)

7.51 (1.8)

    Height, cm

75.1 (9.4)

77.1 (10.2)

76.1 (9.8)

    MUAC, mm

11.8 (0.9)

11.9 (0.7)

11.9 (0.8)

    Weight-for-height, z-score

-3.1 (0.7)

-3.0 9 (0.5)

-3.0 (0.6)

    Weight-for-age, z-score

-.3.4 (0.9)

-3.3 (0.8)

-3.3 (0.8)

    Height-for-age, z-score

-2.4 (1;3)

-2.2 (1.5)

-2.3 (1.4)

Anaemia

Haemoglobin, g/dl

10.7 (1.7)

10.8 (1.3)

10.8 (1.5)

Anaemia, % (n)1*

62.3 (33)

51.0 (25)

56.9 (58)

Haemoglobinopathy**

Hb normal, % (n)

58.2 (32)

71.4 (35)

64.4 (67)

Hb disorders, % (n)

42.8 (23)

28.6 (14)

35.6 (37)

  HbE-Heterozygote, % (n)

27.3 (15)

14.3 (7)

21.2 (22)

  HbE-Heterozygote/β-thalassemia, % (n)

3.6 (2)

4.1 (2)

3.9 (4)

  HbE-Homozygote, % (n)

5.5 (3)

4.1 (1)

4.8 (5)

  β-thalassemia major, % (n)

1.8 (1)

0.0 (0)

1.0 (1)

  α-thalassemia, % (n)

0.0 (0)

2.0 (1)

1.0 (1)

  α-thalassemia/Harbour spring, % (n)

0.0 (0)

2.0 (1)

1.0 (1)

  Hb-Other, % (n)

3.6 (2)

2.0 (1)

3.0 (3)

Iron status*

Ferritin, µg/L

40.3 (30.0)

41.4 (28.1)

40.8 (28.9)

sTfR, mg/L

10.2 (7.4)

10.1 (9.0)

10.1 (8.2)

BI, mg/kg

3.5 (4.6)

4.1. (3.8)

3.8 (4.2)

Iron deficiency, % (n)

19.3 (11)

5.7 (3)

12.7 (14)

Low BI, % (n)

17.5 (10)

11.3 (6)

14.5 (16)

Vitamin A status*

RBP, µmol/L

1.2 (0.5)

1.4 (0.6)

1.3 (0.6)

Marginal vitamin A status, % (n)

38.6 (22)

24.5 (13)

31.8 (35)

Vitamin A deficiency, % (n)

7 (4)

3.8 (2)

5.5 (6)

Inflammation status*

CRP, mg/L

4.0 (10.8)

3.1 (5.4)

3.6 (8.1)

CRP, >5 mg/L, % (n)

12.3 (7)

18.9 (10)

15.5 (17)

AGP, g/L

1.1 (0.9)

1.0 (0.6)

1.1 (0.8)

AGP, >1 g/L, % (n)

38.6 (22)

39.6 (21)

39.1 (43)

Data are reported as percentages (n) or mean (SD). 1Anaemia cutoff is age and sex-dependent. Abbreviations: MUAC, mid-upper-arm circumference; sTfR, soluble transferrin receptor; BI, body store; CRP, C-reactive protein; AGP, α1-acid glycoprotein. * No micronutrient data for one child (standard RUTF) and three children (local RUTF), hence a total of 4 children. ** Missing data from 10 children (n = 3 standard RUTF and n = 7 local RUTF).

R1.C6: How was iron deficiency defined?

R1.C6 Authors reply: Iron deficiency was defined as FER<12 μL for children <60 months of age.

For clarity, it has been revised to: "No iron stores; hence, iron deficiency was defined as FER<12 μL for children <60 months of age" (line 177).

R1.C7: Most of the variables (iron indices) seem not normally distributed, consider providing median and quartiles.

R1.C7 Authors reply: Biological data does indeed tend to be non-normal distributed. Checking the normality for ferritin at baseline and endline shows a fairly normal data distribution. We also checked differences among the RUTF groups with non-parametrical Mann-Whitney-U test, and no different between the groups were found. After consideration, the authors decided to keep using mean and SD; hence the authors find that using means and SD’s are sufficient in this case. If the reviewer insists on changing to median and quartiles, the authors are willing to accommodate the change.

R1.C8: Is it possible that nutrient interactions (for example, between iron and zinc) or malnutrition enteropathy reduced the effect of the RUTF? Add in discussion.

R1.C8 Authors reply:

We believe indeed that gut (or systemic) inflammation affects the bioavailability of for example iron and vitamin A, but data are scare. We do not believe zinc – iron interactions have played a role, as interactions between iron and zinc have been reported to be low in zinc:iron ratios of 1:5 to 5:1.

To reflect the reviewer's comment, the authors have added the following to the discussion:

Malnutrition enteropathy:

"Small intestinal inflammation (environmental enteric dysfunction and environmental enteropathy) is widespread in low- and middle-income countries are strongly associated with malnutrition, for example, stunting, and wasting [49–53] and hence environmental enteric dysfunction and enteropathy are likely to influence and limit the effectiveness of fortified foods such as RUTFs in improving micronutrient status in malnourished children" (line 334-338).

Micronutrient interactions:

"Both the local and standard RUTF contains ~2 mg more zinc than iron, so it could be speculated that the low increase in iron in this trial was caused by micronutrient interactions between different nutrients resulting in inhibited uptake of certain nutrients, for example, zinc and iron compete during intestinal absorption when they are ingested simultaneously, which is the case when children are treated with RUTFs [54]. Excess zinc intake has been found to inhibit zinc absorption and vice versa [55,56]. However, the evidence is conflicting and discussions on definitions of interactions between micronutrients [57]." (lines 340-347).

R1.C9: Iron intake has shown to be detrimental for growth (Bo Lönnerdal, Excess iron intake as a factor in growth, infections, and development of infants and young children, The American Journal of Clinical Nutrition, Volume 106, Issue suppl_6, December 2017, Pages 1681S–1687S), should we be cautious in administering iron in iron replete children. Discuss.

R1.C9 Authors reply: To reflect the reviewer's comments, the authors have added the following to the discussion:

"Another concern with iron is that iron is a pro-oxidative element, which can have negative effects on biological systems at even moderate dosages, for example, beyond altering the gut microbiota, there are some reports on decreased growth (linear and weight), increase illness and mentioned earlier interactions with other trace elements such as zinc and copper and impaired cognitive and motor development [59]. (lines 364-369).

R1.C10: Add sample size as a limitation as the original sample size calculation was not targeted to the present study outcome. 

R1.C10 Authors reply: The following was added to section 4.1. Strengths and limitations line 402-404:

"It's important to note that the study presents secondary outcomes, so the sample size calculation was not calculated to assess the difference in micronutrient status:

Minor comments from reviewer 1:

R1.C11: 1. Table 2: Food 'ratio' to 'ration'

R1.C11 Authors reply: In Table 2: Food 'ratio' have been revised to food 'ration'. Thank you for catching that.

R1.C12: 2. Line 193 close brackets for SD.

R1.C12 Authors reply: The authors understand that it appears like the definition for SD was not closed because of a line change. Below is the explanation:

In line 193 (now 199), the following sentence is written: "Descriptive statistics are presented as percentages (n) or mean (standard deviation," the line continues to line 200 with "SD). The difference between…". The full sentence is then: "Descriptive statistics are presented as percentages (n) or mean (standard deviation, SD)", having closed brackets after defining abbreviations for SD.

R1.C13: 3. Add reference for lines 317 – 321.

R1.C13 Authors reply: Lines 317 – 321 (now 325-330) refer to an ad-hoc analysis of children with high compliance for the present study; however, results are not shown. The following have been added/revised for clarity:

"During the present study, compliance with RUTF intake was lower than expected and desired [47], which influenced the intake of micronutrients and hence the possible increase in micronutrient status between admission and discharge. In an ad-hoc analysis of the data from the present study (data not presented), even in the small sample of children with high compliance, no indication or significant impact on micronutrient status was found. Hence, it appears that the RUTFs provided insufficient micronutrients to improve micronutrient status, confirming similar observations by Kangas et al. [29]".

Reviewer 2 Report

Thank you for the opportunity to review this manuscript. A deficiency of vitamin A is associated with significant morbidity and mortality from common childhood infections and is the world's leading preventable cause of childhood blindness.VAD also causes substantial morbidity. It is among the “top ten” health problems contributing to the global disease burden.

 This manuscript is interesting, but it has gaps. First of all, the introduction could be implemented with specific information. What causes vitamin A deficiency? What are the main ways to combat VAD? In addition, it's essential to write focus on specific properties of vitamin A: viral activity ( Sinopoli, A., Caminada, S., Isonne, C., Santoro, M. M., & Baccolini, V. (2022). What are the effects of vitamin A oral supplementation in the prevention and management of viral infections? A systematic review of randomized clinical trials. Nutrients, 14(19), 4081), antioxidant activity ( Palace, V. P., Khaper, N., Qin, Q., & Singal, P. K. (1999). Antioxidant potentials of vitamin A and carotenoids and their relevance to heart disease. Free Radical Biology and Medicine, 26(5-6), 746-76), in metabolic disease ( Blaner, W. S. (2019). Vitamin A signaling and homeostasis in obesity, diabetes, and metabolic disorders. Pharmacology & therapeutics, 197, 153-178), etc...

In the methods, it's necessary to specify that it is a retrospective study and the follow-up time. Why was mortality not assessed? Please,I ask the authors to provide an explanation.

I suggest implementing the discussion with the foregoing about the role of vitamin A 

Author Response

Reviewers' report, revisions and response to reviewers' reports                                                         

Dear Reviewers,

A sincere thanks for spending your valuable time reviewing our manuscript. The concerns have been addressed in reply to your comments below. The comments and suggestions have been ordered as R1 (reviewer 1) and C# (comment no) (example R1.C1). The revisions in the text are marked using the" Track Changes" function in MS Word.

Reviewer 2's report and authors' comments/revisions

Open Review

English language and style

( ) English very difficult to understand/incomprehensible
( ) Extensive editing of English language and style required
( ) Moderate English changes required
(x) English language and style are fine/minor spell check required
( ) I don't feel qualified to judge about the English language and style

Yes

Can be improved

Must be improved

Not applicable

Does the introduction provide sufficient background and include all relevant references?

( )

( )

(x)

( )

Are all the cited references relevant to the research?

( )

( )

(x)

( )

Is the research design appropriate?

( )

(x)

( )

( )

Are the methods adequately described?

( )

(x)

( )

( )

Are the results clearly presented?

(x)

( )

( )

( )

Are the conclusions supported by the results?

( )

(x)

( )

( )

Reviewer comments to Authors:

R2.C1: Thank you for the opportunity to review this manuscript. A deficiency of vitamin A is associated with significant morbidity and mortality from common childhood infections and is the world's leading preventable cause of childhood blindness. VAD also causes substantial morbidity. It is among the "top ten" health problems contributing to the global disease burden.

R2.C1 Authors reply: Thank you very much for your interest and for your time to review this manuscript. We agree that vitamin A deficiency is a public health problem contributing to the global disease burden associated with significant morbidity and mortality. The challenges with children who often suffer from micronutrient deficiencies combined with being severely malnourished demand a broadened perspective on managing micronutrient deficiencies in children with severe acute malnutrition at a 5-20-fold higher risk of morbidity than normal weight children (World Health Organization (WHO). Guideline: Updates on the
Management of Severe Acute Malnutrition in Infants and Children.
WHO; 2013).

R2.C2. This manuscript is interesting, but it has gaps. First of all, the introduction could be implemented with specific information. What causes vitamin A deficiency?

R2.C2 Authors reply: The authors appreciate and acknowledge the reviewer's comments. In the introduction, line 41-44 writes: "SAM contributes significantly to childhood mortality and morbidity [1,2] and is associated with micronutrient deficiencies, anaemia and infections [3–9]. Micronutrient deficiencies, including iron, zinc and vitamin A, increase mortality risk and contribute to a high prevalence of infection, deprived growth and sub-optimal cognitive development [10]," indicating the association between SAM and micronutrient deficiencies.

Please note that our paper addresses a very specific topic: vitamin A and iron deficiency in children with severe acute malnutrition, and whether current treatment with RUTF is sufficient to address this. We are not elaborating in our paper on the wider context of vitamin A deficiency in populations.

R2.C3. What are the main ways to combat VAD? In addition, it's essential to write focus on specific properties of vitamin A: viral activity ( Sinopoli, A., Caminada, S., Isonne, C., Santoro, M. M., & Baccolini, V. (2022).

R2.C3 Authors reply: Please see our comments above. The main focus of the current paper is whether vitamin A and iron status are improved during treatment for severe acute malnutrition. Children with SAM's main pathway to combat and recover from VAD is through ready-to-use therapeutic foods (RUTFs), and these were supposedly designed to supply sufficient micronutrients, including vitamin A, for children to regain their weight and micronutrient deficiencies. The current study and Suvi et al. show that this might not be the reality and that RUTFs contain insufficient amounts of vitamin A and iron to recover from their deficiencies fully.

Combatting VAD in populations is another topic, indeed a very interesting one, but outside the scope of the present paper.

R2.C4. What are the effects of vitamin A oral supplementation in the prevention and management of viral infections? A systematic review of randomized clinical trials. Nutrients, 14(19), 4081), antioxidant activity ( Palace, V. P., Khaper, N., Qin, Q., & Singal, P. K. (1999). Antioxidant potentials of vitamin A and carotenoids and their relevance to heart disease. Free Radical Biology and Medicine, 26(5-6), 746-76), in metabolic disease ( Blaner, W. S. (2019). Vitamin A signaling and homeostasis in obesity, diabetes, and metabolic disorders. Pharmacology & therapeutics, 197, 153-178), etc...

R2.C4 Authors reply: Please see our reply above. The authors find that adding these topics is beyond the scope of the present manuscript, also thinking that the same review would be required as a minimum for iron.

R2.C5. In the methods, it's necessary to specify that it is a retrospective study and the follow-up time.

R2.C5 Authors reply: The authors disagree with that the study as a retrospective study. The authors understand a retrospective study to be one where participants already have a known disease or outcome, e.g., diabetes. A retrospective study looks back into the past to try and determine why the participants have diabetes and when they might have been exposed (https://www.statisticshowto.com/cohort-study/). The manuscript presents data on a prospective, randomized controlled trial using data on micronutrient status at admission and after eight weeks of treatment (Lines 84-85). The trial's primary outcome was weight gain, and these results have been published elsewhere: https://pubmed.ncbi.nlm.nih.gov/30012981/. The children were treated for eight weeks with RUTFs, comparing a local product with a standard product (active control). If the reviewer and Editor insist that adding this information will benefit the manuscript and its objective, the authors will include it in the 2nd round of revisions.

R2.C6. Why was mortality not assessed? Please, I ask the authors to provide an explanation.

R2.C6 Authors reply: Figure 1, lines 223-242, shows that two children receiving BP-100 (the standard RUTF) passed away during the treatment. For clarity, the following was added to lines 222-224: Two children allocated to the standard RUTF, unfortunately, passed away during the study because of progressed HIV infection (one child) and pulmonary tuberculosis (second child).

R2.C7. I suggest implementing the discussion with the foregoing about the role of vitamin A.

R2.C7 Authors reply: The foregoing discussion about vitamin A's role is very important, but the authors find that it goes beyond the scope of the manuscript.

Round 2

Reviewer 2 Report

This version is ok for me.